# Learning Shape Reconstruction from Sparse Measurements with Neural Implicit Functions

**Tamaz Amiranashvili**[1,2,3]                                 TAMAZ.AMIRANASHVILI@UZH.CH
**David Lüdke**[2]                                                              LUEDKE@ZIB.DE
**Hongwei Li**[1,3]                                                       HONGWEI.LI@TUM.DE
**Bjoern Menze**[1,3]                                               BJOERN.MENZE@UZH.CH
**Stefan Zachow**[2]                                                          ZACHOW@ZIB.DE

[1] *Department of Quantitative Biomedicine, University of Zurich, Zurich, Switzerland*
[2] *Visual and Data-Centric Computing, Zuse Institute Berlin, Berlin, Germany*
[3] *Department of Computer Science, Technical University of Munich, Munich, Germany*

**Editors:** Under Review for MIDL 2022

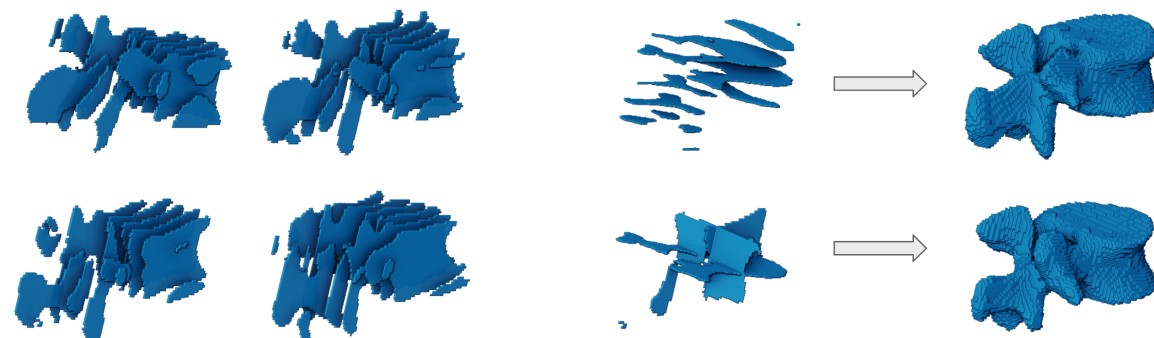

(a) Sagittal training data examples          (b) Reconstructions from different inputs

Figure 1: Our model can be trained on sparse, anisotropic segmentations (a) and is able to reconstruct shapes at any target resolution from a variety of sparse input configurations (b). Visualized are actual reconstructions of our model.

## Abstract

Reconstructing anatomical shapes from sparse or partial measurements relies on prior knowledge of shape variations that occur within a given population. Such shape priors are learned from example shapes, obtained by segmenting volumetric medical images. For existing models, the resolution of a learned shape prior is limited to the resolution of the training data. However, in clinical practice, volumetric images are often acquired with highly anisotropic voxel sizes, e.g. to reduce image acquisition time in MRI or radiation exposure in CT imaging. The missing shape information between the slices prohibits existing methods to learn a high-resolution shape prior. We introduce a method for high-resolution shape reconstruction from sparse measurements without relying on high-resolution ground truth for training. Our method is based on neural implicit shape representations and learns

a continuous shape prior only from highly anisotropic segmentations. Furthermore, it is able to learn from shapes with a varying field of view and can reconstruct from various sparse input configurations. We demonstrate its effectiveness on two anatomical structures: vertebra and distal femur, and successfully reconstruct high-resolution shapes from sparse segmentations, using as few as three orthogonal slices.

**Keywords:** shape reconstruction, shape priors, neural implicit shape representations.

## 1. Introduction

In medical imaging, reconstructing shapes of anatomical structures from only a few sparse measurements is required in many contexts, such as reconstruction from segmentations of a few orthogonal slices (Tóthová et al., 2020), or from segmentations of highly anisotropic volumetric images (Turella et al., 2021). Being an ill-posed problem, solving it requires prior knowledge of the distribution of shapes of a given anatomical structure that occur in a population. Various shape models for different shape representations have been developed in the past.

For surface meshes, statistical shape models are widely used in medical shape analysis (Heimann and Meinzer, 2009; Lüthi et al., 2017; Ambellan et al., 2021). They have also been successfully applied for reconstruction tasks (Bernard et al., 2017; Tóthová et al., 2020). Their first limitation is that they require dense point correspondence between training shapes, which is tedious to obtain. Second, they require surface meshes for training, which are hard to extract from volumetric image segmentations, especially if these segmentations are very anisotropic. For voxel-based representations, CNNs have been widely used to learn shape priors and have also been applied for shape reconstruction (Oktay et al., 2017; Cerrolaza et al., 2018; Turella et al., 2021). CNNs can handle varying fields of view and do not require correspondences between training examples, but are fixed to a single discretization and therefore rely on resampling of volumes with variable spacings to a common one. In addition, the reconstruction task (e.g., spacing between slices or acquisition direction of input volumes) has to be known during training in order to generate a realistic distribution of source inputs. Most importantly, however, both approaches have a strong common limitation – their resolution is limited to the resolution of their training data.

Recently, neural implicit functions have gained popularity as shape representations (Mescheder et al., 2019; Park et al., 2019; Chen and Zhang, 2019). In contrast to CNNs, their main advantage is a continuous formulation, allowing sampling shapes at arbitrary resolutions both during training and inference. This property makes implicit functions particularly useful in medical imaging, where volumetric grids with varying spacings and resolutions occur naturally. In particular, anisotropic scans with low resolution in one axis are prevalent in medical imaging to reduce motion artefacts, acquisition time, and radiation exposure. We pose a novel task of utilizing low-resolution, anisotropic volumetric segmentations to learn a shape prior that is able to perform high-resolution shape reconstruction *beyond* the resolution of training data (cf. Figure 1). We demonstrate that implicit functions are able to solve this problem, in contrast to existing CNNs and statistical shape models. While shape reconstruction from sparse inputs with implicit functions has been studied in the computer vision community (Mescheder et al., 2019; Chibane et al., 2020), in contrast to our work, these methods rely on training data in target resolution.

In summary, our contributions are as follows:

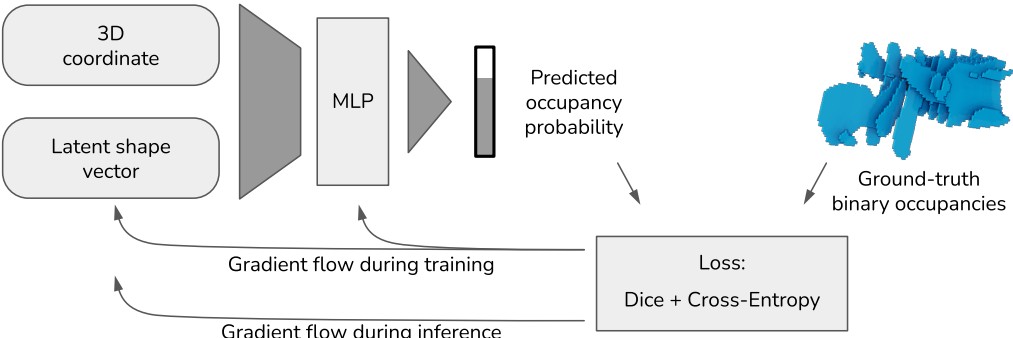

Figure 2: During training, we optimize the MLP as well as individual latent vectors per training shape. During inference, only the latent vector is optimized, based on reconstruction input volume.

1. we propose a method that is able to reconstruct shapes from various sparse measurements, such as anisotropic segmentations with arbitrary orientation and spacing, up to as few as only three orthogonal slices,

2. our method does not rely on any ground-truth in the target resolution – for training and model selection,

3. we evaluate the proposed method quantitatively and qualitatively on two publicly available datasets including two anatomical structures and publish the source code[1].

## 2. Methods

### 2.1. Shape Representation

A 3D shape is modelled as a decision boundary of a binary classifier (Mescheder et al., 2019). That is, for a given continuous 3D coordinate $x \in \mathbb{R}^3$ in the ambient space, such classifier predicts whether the point $x$ lies inside or outside of the given shape. Modelling every shape through its own, independent classifier is not useful, since we aim to learn a shared shape prior for all shapes within a given population. Therefore, a single classifier is used, which is shared among all shapes, but is conditioned on a latent vector $z \in \mathbb{R}^d$ to represent shape variations. The classifier is modelled by a multilayer perceptron (MLP) network $f_\theta$, which takes as input both the 3D coordinate and the latent vector, and returns an *occupancy probability* of this point being inside the target shape:

$$f_\theta : \mathbb{R}^3 \times \mathbb{R}^d \to [0, 1]$$

We choose to predict the occupancy instead of continuous distance fields to the shape boundary, since 3D distance fields cannot be accurately estimated for highly anisotropic ground-truth segmentations. For architecture details of $f_\theta$, please cf. Appendix A.

---

1. https://github.com/menzelab/implicit-shape-reconstruction

## 2.2. Shape Prior Training

For training of the shape prior, we rely on a set of volumetric, voxel-based binary ground-truth segmentations. The continuous shape formulation allows us to use heterogeneous ground-truth segmentations in terms of numbers of voxels, spacings, and anisotropy. To obtain latent representation $z_i$ for every training volume $i$, we adopt the auto-decoder scheme (Bojanowski et al., 2018; Park et al., 2019) (cf. Figure 2). That is, during training, a randomly initialized latent vector $z_i \sim \mathcal{N}(0, 0.1^2)$ is stored for each training shape $i$. The individual latent vectors are optimized jointly with the parameters of the MLP $f_\theta$. For supervision, for every volume $i$ and voxel $j$ we extract the exact voxel coordinates $x_i^j \in \mathbb{R}^3$ based on physical voxel sizes, as well as corresponding ground-truth occupancy values $y_i^j \in \{0, 1\}$. For a batch of randomly chosen voxels, we obtain a set of predicted occupancy values. Their difference to the ground-truth occupancy is minimized through a voxel-based loss function, which is defined as a sum of volume-wise soft Dice score and voxel-wise binary cross-entropy, like in segmentation tasks (Isensee et al., 2021). Additionally, we employ L2 regularization on the latent representations, following (Park et al., 2019). The loss is minimized through ADAM optimizer (Kingma and Ba, 2014).

**Validation**   Although the MLP $f_\theta$ is shared among all training shapes, special care must be taken to prevent overfitting. Depending on the capacity of the model and the number of training epochs, the model may overfit the sparse voxel positions of the training volumes and perform poorly in-between the slices. Hence, we propose a simple, yet effective validation scheme to perform hyperparameter tuning and early stopping. For a small portion of the training shapes, we use some slices for the training of the MLP and corresponding latent vectors, and some slices for validation. This allows us to control how well the network generalizes to positions, which are unseen during the training.

## 2.3. Shape Reconstruction

After training of the MLP, we can now reconstruct a new, previously unseen shape from a given sparse segmentation $\mathcal{S}$. Reconstruction happens in two steps – first, we determine the latent vector corresponding to given input $\mathcal{S}$. Second, we sample the obtained continuous occupancy function at the target resolution to obtain the voxel-based reconstruction.

The latent vector that corresponds to a given observation $\mathcal{S}$ is obtained through gradient descent. We initialize the latent vector from a normal distribution close to zero: $z \sim \mathcal{N}(0, 1\mathrm{e}{-4}^2)$, following (Park et al., 2019). Similarly to the training procedure above, the MLP $f_\theta$ is sampled at positions of voxels from observation $\mathcal{S}$. We minimize the same loss function as during MLP training, except that only the latent vector is optimized this time. The MLP is frozen, since it represents the common shape knowledge that has been learned from the population of training shapes. The loss is minimized w.r.t. all voxel positions and their occupancy values $(x_k, y_k) \in \mathcal{S}$:

$$z_{rec} = \arg\min_z \sum_k L(f_\theta(x_k, z), y_k)$$

In contrast to MLP training above, we find the convergence to be stable, without a need for validation data to control overfitting. The learning rate and number of iterations are

therefore chosen based on the loss convergence. Given the determined latent vector, the resulting continuous occupancy function $f_\theta(\,\cdot\,, z_{rec})$ can be sampled at any desired resolution and spacing to obtain the high-resolution voxel-based reconstruction.

It is worth noting that a single trained MLP can perform reconstructions from a variety of different input observations – such as segmentations in arbitrary directions, spacings and resolutions, i.e. decoupled from the training data resolution. Furthermore, the *target* resolution of the reconstruction is also decoupled from both the training data resolution as well as input resolution, since the acquired representation is continuous.

## 3. Experiments and Results

### 3.1. Datasets

Two publicly available datasets were used for evaluation – spine data from the MICCAI challenge "Large Scale Vertebrae Segmentation Challenge" (*Verse*) (Sekuboyina et al., 2021), and knee data from (Ambellan et al., 2019).

In the *Verse* dataset, we reconstruct shapes of lumbar vertebrae, following clinical motivation in (Turella et al., 2021). The dataset contains healthy and pathological anatomies, such as vertebrae with fractures. For simplicity of experimental design and evaluation, we choose isotropically spaced scans and extract subvolumes of size $128^3$ voxels with an isotropic spacing of $1\,\mathrm{mm}^3$ around each vertebra. This results in 287 volumes, from which we randomly draw 230 volumes for training and validation, and 57 for testing. The training and test sets are separated so that no vertebra from the same patient is in both sets.

In the knee dataset, we reconstruct the shape of the distal femur. The dataset includes healthy and pathological anatomies with varying field of view. All volumes have a resolution of $160 \times 384 \times 384$ voxels and an anisotropic voxel size of $0.7$ mm $\times$ $0.36$ mm $\times$ $0.36$ mm. We randomly choose 254 volumes for training and validation, and 100 volumes for testing.

Both datasets exhibit pose variations – rotational between the vertebrae and translational between femur bones. The data have not been aligned to demonstrate the model's ability to deal with realistic pose variations.

### 3.2. Shape Prior Training

Our goal is to demonstrate that our model can learn a continuous shape prior from sparse data. According to (Turella et al., 2021), it is realistic for volumetric scans to have an anisotropic voxel size of 4.7mm $\times$ 0.6mm $\times$ 0.6mm, which corresponds to approx. 8-fold anisotropy along the sagittal axis. To simulate such anisotropy in our data, we generate training data from high-resolution ground-truth volumes by taking every 8-th slice on the sagittal axis, skipping the slices in-between. The positions of the chosen slices are randomly selected per patient. This selection happens only once per patient and is fixed during training. This assures that we have only a single set of sparse slices per patient, which represents a realistic, but challenging training scenario. Please cf. Appendix A for architecture and Appendix B for optimization details.

### 3.3. Reconstruction Performance

In this section, we evaluate our model on various reconstruction tasks and compare the results to baseline methods. We emphasize that the same trained proposed model per dataset was used for all reconstruction experiments below.

**Baselines** To the best of our knowledge, there are no prior data-driven methods, which can reconstruct shapes from sparse segmentations without relying on ground truth in the target resolution for training. Therefore, we compare our results quantitatively to trilinear and cubic B-spline interpolation of sparse input segmentations, thresholded at 0.5 to obtain binary labels. To put our results additionally into context, we also train a ReconNet from (Turella et al., 2021). In contrast to our method, ReconNet cannot be trained on sparse segmentations. Therefore, it is trained on high-resolution ground-truth data, providing an upper bound to our method. We omit the VAE component from (Turella et al., 2021), since we only focus on the reconstruction task.

**Metrics and Evaluation** To quantitatively compare our reconstructions to ground-truth, we sample the resulting continuous occupancy function at the same volumetric grid as our original ground-truth volumes. This allows us to precisely compare the two volumes with established metrics – average surface distance, 95-percentile Hausdorff distance and Dice score. Note that we only use the ground truth in target resolution at test time to evaluate our method.

**Reconstruction from Sagittal Slices** First, we evaluate our method on the reconstruction task from anisotropic segmentations with 8-fold anisotropy in the sagittal direction. The input distribution is therefore similar to the one in training examples. Quantitative results for both datasets on corresponding test sets can be found in Table 1. All three metrics show a consistent picture on both datasets. The femur dataset exhibits better accuracy overall since it's a much smoother and less intricate structure than the lumbar vertebrae, which makes the reconstruction task easier. Our method outperforms both interpolation methods by a large margin. However, if high-resolution training data is available and the reconstruction task is fixed and known beforehand, ReconNet delivers even more accurate results. It is worth noting that the femur volumes were too large for the ReconNet to even perform inference on a single example on a 32GB GPU. Therefore, we had to resort to training on random crops and fuse them at inference time. In contrast, our model is very flexible w.r.t. batch size, since all coordinates are inferred independently, allowing arbitrary batch sizes. For more results on thick slice segmentations, cf. Appendix E.

**Reconstruction from Axial and Coronal Slices** Second, we reconstruct shapes from axial as well as coronal sparse slices with 8-fold anisotropy (cf. Table 2). Our results are similar to reconstruction from sagittal slices (cf. Table 1), which shows that our model generalizes well to new, previously unseen spacings in the input. See Figures 1 and 3 for qualitative results.

**Reconstruction from Orthogonal Slices** Our last reconstruction setting is reconstruction from three orthogonal slices (cf. Table 3). While the metrics drop compared to the two reconstruction tasks above, they are still in a reasonable range. First, it showcases the ability of our model to reconstruct shapes from very sparse inputs, far from the distribution

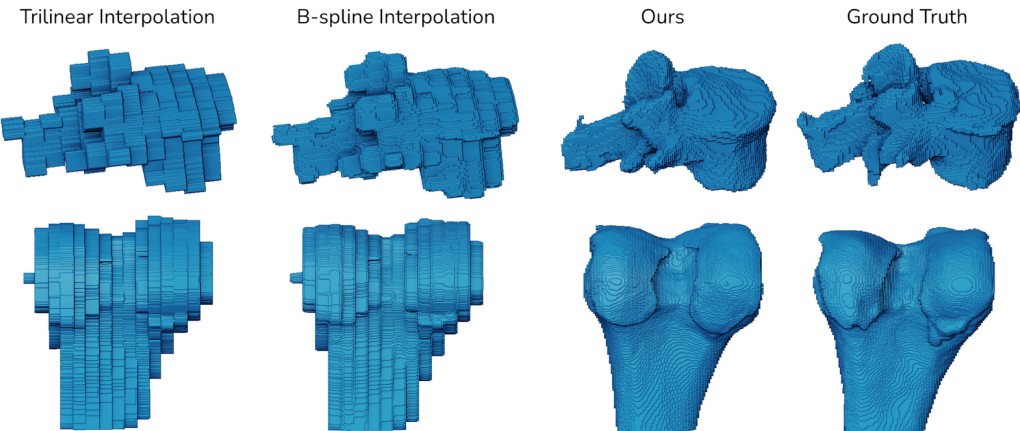

Figure 3: Reconstruction results for coronal input slices of the vertebra and sagittal input slices of the femur.

of inputs seen during training. Second, it highlights that our model does not smoothly interpolate any given input observation individually, but has indeed learned a population-wide prior. In this example, no interpolation technique would have been able to extrapolate beyond the input observations without a data-driven shape prior, like our model does. See Figure 1 as well as Figures A1 and A2 in the Appendix for qualitative results.

### 3.4. Generative Model

Our model follows a generative approach and therefore allows sampling shapes from the learned latent representation. To evaluate the validity of the emerging latent space qualitatively, we generate a shape represented by the mean of the learned latent vectors from the training set. Figure A3 in the Appendix shows our learned mean shapes of vertebra and distal femur, sampled at very high resolution – beyond the resolution of the training data in all axes. We thereby qualitatively show that our model can learn and produce a plausible mean shape from sparse measurements.

### 4. Conclusion

We have posed a novel task of reconstructing high-resolution shapes from sparse measurements without relying on high-resolution data for training. To the best of our knowledge, this task has not been attempted to be solved before. We show that neural implicit functions are an effective tool in this context. The reconstructions are good on average, even though small, patient-specific details may be missing. In future work, uncertainty quantification would make the reconstructions even more reliable. Besides the good reconstruction capabilities, our model provides a common, compact latent representation, allowing unification of heterogeneous segmentations in a joint shape prior. This work therefore opens up possibilities to study the emerging latent space for statistical shape analysis, as well as embedding the shape prior into other ill-posed segmentation tasks in the future work.

Table 1: Test set accuracy for vertebra and distal femur reconstruction from input segmentations with 8-fold anisotropy in sagittal axis. Note that in contrast to our method, ReconNet has been trained on full-resolution data and, therefore, represents an *upper bound* for our method. The metrics are average surface distance (ASD), 95-percentile Hausdorff distance (HD95), and Dice similarity coefficient (DSC).

| *Vertebra* | ASD [mm] ↓ | HD95 [mm] ↓ | DSC ↑ |
|---|---|---|---|
| Trilinear Interpolation | $0.69 \pm 0.08$ | $2.52 \pm 0.32$ | $0.89 \pm 0.01$ |
| B-spline Interpolation | $0.63 \pm 0.09$ | $2.52 \pm 0.53$ | $0.89 \pm 0.01$ |
| Ours | $\mathbf{0.48 \pm 0.13}$ | $\mathbf{2.20 \pm 0.93}$ | $\mathbf{0.92 \pm 0.01}$ |
| ReconNet | $0.23 \pm 0.05$ | $1.17 \pm 0.31$ | $0.95 \pm 0.01$ |
| *Femur* | | | |
| Trilinear Interpolation | $0.52 \pm 0.02$ | $1.86 \pm 0.12$ | $0.96 \pm 0.00$ |
| B-spline Interpolation | $0.49 \pm 0.02$ | $1.74 \pm 0.14$ | $0.96 \pm 0.00$ |
| Ours | $\mathbf{0.25 \pm 0.07}$ | $\mathbf{0.92 \pm 0.27}$ | $\mathbf{0.98 \pm 0.00}$ |
| ReconNet | $0.14 \pm 0.03$ | $0.74 \pm 0.09$ | $0.99 \pm 0.00$ |

Table 2: Test set accuracy for vertebra and distal femur reconstruction from input segmentations with 8-fold anisotropy in all axes. These results highlight the model's ability to generalize to anisotropy in various directions. The metrics are average surface distance (ASD), 95-percentile Hausdorff distance (HD95), and Dice similarity coefficient (DSC).

| *Vertebra* | ASD [mm] ↓ | HD95 [mm] ↓ | DSC ↑ |
|---|---|---|---|
| Sagittal | $0.48 \pm 0.13$ | $2.20 \pm 0.93$ | $0.92 \pm 0.01$ |
| Coronal | $0.47 \pm 0.13$ | $2.12 \pm 0.86$ | $0.92 \pm 0.02$ |
| Axial | $0.52 \pm 0.15$ | $2.22 \pm 0.86$ | $0.91 \pm 0.02$ |
| *Femur* | | | |
| Sagittal | $0.25 \pm 0.07$ | $0.92 \pm 0.27$ | $0.98 \pm 0.00$ |
| Coronal | $0.22 \pm 0.06$ | $0.84 \pm 0.21$ | $0.98 \pm 0.00$ |
| Axial | $0.22 \pm 0.06$ | $0.83 \pm 0.21$ | $0.98 \pm 0.00$ |

Table 3: Test set accuracy for vertebra and distal femur reconstruction from three orthogonal segmentation slices. The metrics are average surface distance (ASD), 95-percentile Hausdorff distance (HD95), and Dice similarity coefficient (DSC).

| *Vertebra* | ASD [mm] ↓ | HD95 [mm] ↓ | DSC ↑ |
|---|---|---|---|
| Ours | $0.89 \pm 0.29$ | $3.77 \pm 1.44$ | $0.87 \pm 0.03$ |
| *Femur* | | | |
| Ours | $0.59 \pm 0.15$ | $2.09 \pm 0.50$ | $0.96 \pm 0.01$ |

## Acknowledgments

We thank Dennis Madsen, Alexander Tack and Anjany Sekuboyina for stimulating discussions and useful input.

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

## Appendix A. Architecture Details

For the occupancy prediction, our latent vectors have the dimensionality 128 for both datasets. Our MLP has 8 linear layers with 128 dimensions each, with skip-connections and ReLU non-linearities in-between. The coordinates are concatenated with the latent vector at the MLP input, as well as with intermediate activations at layer 4, similar to (Park et al., 2019).

## Appendix B. Optimization Details

For training of the MLP, we use a learning rate of 0.001 and batches of 8 volumes. For each epoch, we randomly select $64^3$ voxels per example volume within a batch. The latent

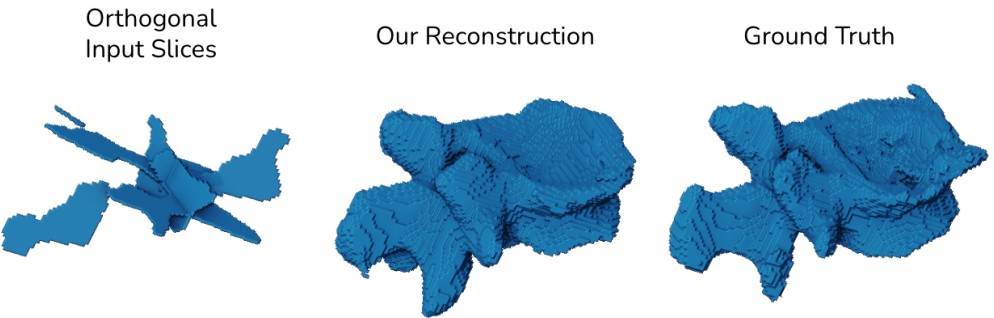

Figure A1: Reconstruction of a pathological vertebra from three orthogonal slices.

vector regularization coefficient was set to 1e-4. Models for both datasets were trained in under 12 hours each on an Nvidia V100 32GB GPU.

For optimization of the latent vector during reconstruction, we use a learning rate of 0.001 and latent vector regularization coefficient of 1e-4. Instead of using stochastic gradient descent with minibatches, we use all available voxels (per shape) in each gradient step. Number of optimization steps represents a trade-off between reconstruction speed and accuracy. For vertebra, we optimize for 1200 steps for anisotropic reconstructions and for 2000 steps for orthogonal reconstructions. For femur, we optimize until the rounded training dice score stops increasing for 5 blocks of 100 steps in a row to speed up the optimization.

Time required for latent vector optimization depends on the number of voxels present in the sparse reconstruction input. Furthermore, we found that some shapes converge after fewer steps than others. On average, latent vector optimization for sagittal anisotropic inputs took ∼1min and ∼2.5min per shape for vertebra and femur correspondingly on a consumer-grade Nvidia RTX 3090 24GB GPU. Consequent high-resolution sampling is fast with ∼0.1s per vertebra, and ∼1s per distal femur.

## Appendix C. Learned Mean Shape

Since our model is a generative one, we are able to sample the shape at the mean of training latent vectors. The resulting shapes look like mean shapes of the population. Figure A3 in the Appendix shows these shapes, sampled at very high resolution – beyond the resolution of the training data in all axes.

## Appendix D. Comparison to Image Segmentation Performance

To put the achieved results into a larger context, we provide metrics for state of the art segmentation methods on full-resolution image data. For lumbar vertebra segmentation from CT images, winner method in (Sekuboyina et al., 2021) achieved a Dice score of ∼0.92 (ASD and HD95 metrics not available). For the distal femur segmentation from MRI images, (Ambellan et al., 2019) achieved an average surface distance of $0.17 \pm 0.05$mm and Dice score of $0.986 \pm 0.003$ (HD95 metric not available).

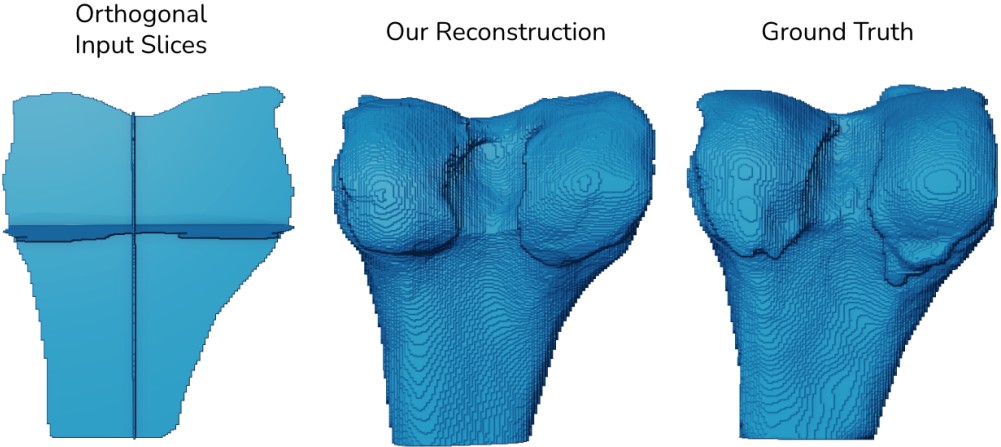

Figure A2: Reconstruction of a pathological femur from three orthogonal slices.

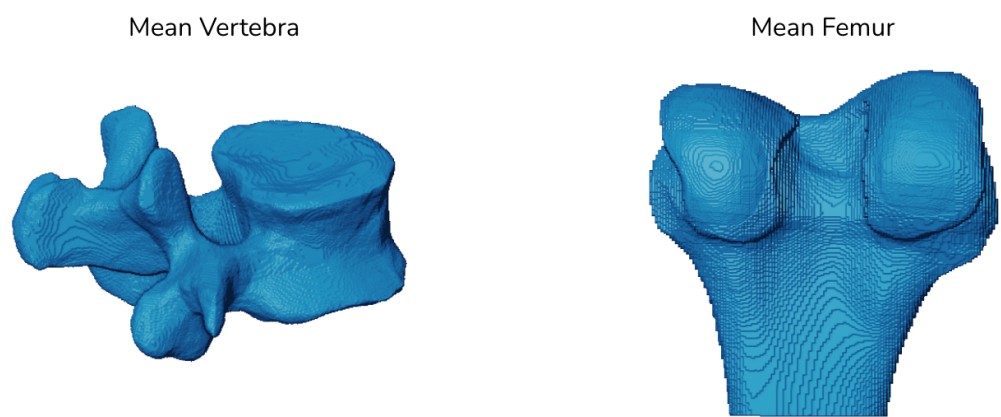

Figure A3: Learned mean shape of our generative model sampled at high resolution represented by the mean of latent representations in the training set.

Table A1: Test set accuracy for vertebra and distal femur reconstruction from input segmentations with thick slices and 8-fold anisotropy in sagittal axis. Note that in contrast to our method, ReconNet has been trained on full-resolution data and, therefore, represents an *upper bound* for our method. The metrics are average surface distance (ASD), 95-percentile Hausdorff distance (HD95), and Dice similarity coefficient (DSC).

| *Vertebra* | ASD [mm] ↓ | HD95 [mm] ↓ | DSC ↑ |
|---|---|---|---|
| Trilinear Interpolation | $0.69 \pm 0.07$ | $2.80 \pm 0.26$ | $0.89 \pm 0.01$ |
| B-spline Interpolation | $0.63 \pm 0.08$ | $2.36 \pm 0.33$ | $0.89 \pm 0.01$ |
| Ours | $\mathbf{0.52 \pm 0.10}$ | $\mathbf{2.08 \pm 0.62}$ | $\mathbf{0.91 \pm 0.01}$ |
| ReconNet | $0.22 \pm 0.04$ | $1.12 \pm 0.23$ | $0.96 \pm 0.01$ |
| *Femur* | | | |
| Trilinear Interpolation | $0.54 \pm 0.02$ | $2.04 \pm 0.09$ | $0.96 \pm 0.00$ |
| B-spline Interpolation | $0.52 \pm 0.03$ | $2.00 \pm 0.13$ | $0.96 \pm 0.00$ |
| Ours | $\mathbf{0.31 \pm 0.05}$ | $\mathbf{1.06 \pm 0.22}$ | $\mathbf{0.98 \pm 0.00}$ |
| ReconNet | $0.14 \pm 0.02$ | $0.73 \pm 0.06$ | $0.99 \pm 0.00$ |

## Appendix E. Thick Slice Performance

In clinical practice it is also common to have thick slices, i.e. slices with large distance, which average the measurements in their surroundings. We simulate such scenario by averaging and thresholding chunks of slices for both training data and reconstruction inputs. Results can be seen in Table A1. We observe that our method has a slight drop in performance compared to thin slices, while the upper bound method (ReconNet) improves a little. This is probably due to the fact that our model did not see precise, high-resolution thin slices in training, making it harder to build a high-resolution prior. ReconNet, on the other hand, benefits from added information in each thick slice about its surroundings. This information can be effectively utilized, since ReconNet still has high-resolution, thin slices as ground-truth during training.

