# OpenReview forum: "Learning Shape Reconstruction from Sparse Measurements with Neural Implicit Functions"
_MIDL.io/2022/Conference — MIDL 2022_

### Official Review · Reviewer_v7Hk · 2022-01-23

**Confidence:** 4
**Preliminary Rating:** 4
**Recommendation:** Oral

**Summary:**

This paper proposes a new method for inferring and upsampling shape representations that can work with anisotropic, or sparsely sampled, segmentation maps. The approach is based on creating a neural implicit representation of shape, where a shape prior is learned from a training dataset (with the same level of anisotropy/sparseness), but following inference shapes can be generated at arbitrary resolutions. The experiments are conducted on femur or vertebrae segmentation datasets and demonstrate the capability to infer reasonable high resolution shapes from lower-resolution data.

**Strengths:**

Building a shape model, that can learn from data with different spatial sampling patterns is clearly a useful development, demonstrating the applicability of these types of shape models to a wider variety of data.

The paper is generally clearly written, and the idea is straightforward and well explained. Figure 2 is a useful illustration of the approach.

The quantitative results indicate a substantial improvement of performance over a simple baseline, and results that are not too far away from methods trained on higher resolution data, which may not always be available.

**Weaknesses:**

At points the authors refer to “thick” slices, but I don’t think the experiments show this. Rather, the experiments cover thin slices with a large gap. It would have interesting to compare the performance against thick slices (e.g. where the 8 slices are averaged) - I’m not sure which of these scenarios is more common in practice but both seem plausible. Importantly, the language of the paper (to me) indicates something different to what is shown in the results.

Why was trilinear your baseline of choice rather than standard higher level interpolators such as cubic splines, or even sinc?

The authors should include the details of their inference procedure in an appendix, elaborating on “Parameters, such as learning rate and number of optimization steps, can be chosen directly based on the training loss”.

**Deanonymize Review:**

no

**Detailed Comments:**

It would have been nice to see how the size of the latent vector was chosen, and the effect of this choice on the results.

Was any initial alignment performed on the data? If so, was this before/after downsampling. Was any augmentation used during train time?

Grammatical mistake in sec 2.1
“Modelling every shape through its own independent classifier is not meaningful, however, since we aim to learn a shared shape prior for all shapes within a given population. “

**Final Rating After The Rebuttal:**

5: Strong Accept

**Justification Of The Final Rating:**

The authors did a good job of taking comments from the reviewers on board, and have incorporated several of these into the current revision. I think this paper will made a good additional to MIDL, so I'm upping my recommendation to strong accept.

**Paper Type:**

methodological development

**Questions To Address In The Rebuttal:**

Showing training runs and/or inference from data with thick slices rather than sparsely sampled slices would be helpful to quantify the differences between these scenarios.
More details regarding the inference procedure are required for the method to be reproducible.
A stronger baseline using cubic spline interpolation should be compared against.

**Special Issue:**

no

---

### Official Review · Reviewer_wDnB · 2022-01-26

**Confidence:** 4
**Preliminary Rating:** 4
**Recommendation:** Poster

**Summary:**

The paper adapts an existing idea of reconstructing 3D shapes with an auto-decoder based generative neural implicit shape function, to reconstruct volumetric shapes of high resolution in medical images using 3D ground truth shape segmentation which may only be available in low resolution during training. The paper motivates the application by highlighting the fact that 3D medical images are often anisotropic with a relatively low resolution across one direction, and hence high-resolution ground truth shapes (manual segmentation) cannot be available for training networks. The method is evaluated in two different datasets of vertebra and femur. The results show that once the network is trained for a particular shape using enough data, it could generalize to reconstruct shape for new data with very sparse annotations such as just three orthogonal slices.

**Strengths:**

The paper is a good adaptation of the shape completion framework introduced in Park et al 2019 to medical imaging data. The paper does well in trying to leverage the fact that although anisotropy and low resolution can result in losing details of anatomy for a particular subject, the information regarding these details may be assumed to be available when considering the whole dataset as the slices across the low-resolution direction can be expected to sample slightly different parts of the anatomy for different subjects.

The ability to reconstruct shapes at a higher resolution than the resolution at which GT data is available is attractive.

Choosing two different datasets of two different anatomies is useful to evaluate the method’s performance for shapes having different levels of detail.

**Weaknesses:**

Iterative optimization of the latent code needs to be performed during inference. Moreover, for an image with N voxels, it seems that the network needs to be run for forward-pass N times (E.g. For 200^3 image, 200^3 forward passes for each of the voxel coordinates?). It would be better to discuss briefly how much impact does this have on the speed during inference and whether the required time is reasonable for the potential target clinical use case.

**Deanonymize Review:**

no

**Detailed Comments:**

Park et al 2019 use a canonical pose and learn priors for different shapes in the same latent code. However, in the proposed method it seems that the orientation and positioning are changing while the shape more or less remains the same (same anatomy) albeit changes across the training population. Would the results of the method improve if the images are initially aligned and brought to the same space? Also, are the images brought to a common coordinate system in the beginning or not? This is not clear in the paper.

**Final Rating After The Rebuttal:**

5: Strong Accept

**Justification Of The Final Rating:**

The authors have addressed my questions and comments, and have also done well in addressing other reviewers' points in the revision. The paper should be of interest to the MIDL community, hence updating the rating.

**Paper Type:**

both

**Questions To Address In The Rebuttal:**

It would be useful to have a discussion regarding the computational time during inference and whether this could be a limitation in practical application or not.

Clarity on the need (or no need) of bringing the individual shapes of various subjects into a common coordinate system, and if yes how was this done.

**Special Issue:**

no

---

### Official Review · Reviewer_Tfvc · 2022-01-27

**Confidence:** 5
**Preliminary Rating:** 5
**Recommendation:** Oral, Poster

**Summary:**

This paper presents an application of implicit signed distance function learning to reconstruct smooth shape models from low-resolution data in medical imaging. It presents a shape prior modeling framework inspired from a recent work to encode simultaneously mean shape and variability, and validates it on two different anatomical structures sampled at low anisotropic resolution.

**Strengths:**

The paper brings an interesting application of deep learning to estimate shape models, based on classical signed distance function (SDF) modeling. This link between the two is particularly interesting as SDFs are powerful models for shape analysis in medical imaging, but resolution and data quality can be a major limitation. The testing of different sub-sampling scenarios on two different anatomical shapes shows the merits of the method in a challenging situation. The article is generally clear and well organized, and many details are provided. Overall, the paper presents a new, interesting solution to a challenging problem with wide appeal for medical imaging applications.

**Weaknesses:**

The main unanswered question here is how different is the method from the original work of Park et al., 2019. The application is clearly different, but the paper is a bit vague in which differences there are in the architecture and/or learning technique. Overall, the deep learning method is more sketched than described, and a more precise presentation of the algorithm would have been beneficial.

Besides the experiments on sub-sampling, it would have been interesting to explore more the shape model: could the shape latent space be used as a basis to study shape variations (comparing principal components of the latent space with those of the fully sampled inputs, for instance)? Are the pathological aspects of the two pathological shape examples preserved in the reconstruction? Finally, a discussion of the limitation of the approach, in particular relating to shape complexity, would have been interesting (would this approach work for, say, cortical surfaces? vasculature? would that require a dramatic increase in network size?).

**Deanonymize Review:**

no

**Detailed Comments:**

The size of the latent shape vector should be given for the two applications. More algorithmic details would be good, especially when describing the loss function in section 2.2.

**Final Rating After The Rebuttal:**

5: Strong Accept

**Justification Of The Final Rating:**

The authors answered all my questions. This article presents a potentially exciting avenue for using deep learning in medical shape analysis, and provides concrete example applications for the clinic.

**Paper Type:**

validation/application paper

**Questions To Address In The Rebuttal:**

What are the differences and similarities with Park et al., 2019?

Is the shape prior and latent space obtained directly relevant for statistical shape analysis?

How would the approach need to change for more complex shapes?

**Special Issue:**

yes

---

### Official Review · Reviewer_fsib · 2022-01-27

**Confidence:** 4
**Preliminary Rating:** 4
**Recommendation:** Oral

**Summary:**

The authors propose using Neural Implicit Functions to reconstruct high resolution shapes from low resolution segmentation. The method does not require ground truth of the target resolution. The problem is initially formulated as a classification problem, where given a set of xyz coordinates and a latent vector - which accounts for shape variations - a classifier provides a shape occupancy probability. Performance is compared against baseline approaches, with appropriate metrics. Showing very good reconstruction performance achieved from various views as well as using only three orthogonal slices.

**Strengths:**

- The paper is well organized
- Exhaustive experiments were conducted to investigate the performance of the approach against other baseline approaches.
- The idea leads to very good segmentation quality in the high res space.
- The idea has potential to have an impact in the MSK community, since higher resolution shapes could be extracted from low resolution annotations, making annotation process (manual or automated) cheaper and potentially easier.



**Weaknesses:**


Paper's weaknesses are in the clarity of certain steps.

- I would like the authors to add more details about the shape reconstruction at inference time. My understanding is that the workflow involves a training step with frozen MLP, aimed at defining the current latent vector. This relies on the minimization of the loss function used in training. I lack understanding how a low resolution shape turns into a high resolution shape at test time -since the model has never seen a full shape, could the authors please comment on this? Is it because at test time the occupancy probability is queried for additional coordinates as opposed to what is done in training?

- How was the selection of random 304 volumes made? Why 304? OAI is a peculiar study where each patient was monitored multiple time over the span of many years, The random selection my cause having images from the same patient in different splits - please clarify that proper precautions have been taken to prevent info leakage.

- I think the paper is missing reasoning about the limitations of the approach and how the authors see this to be applied in real world. Do they see it coupled to a segmentation algorithm which acts on low res images, and consequently the approach converts that segmentation to a super resolution shape, or would the authors recommend a practitioner to manually segment 3 orthogonal slices (to guarantee correct starting shape) and the model takes care of the generation of the full shape?



**Deanonymize Review:**

no

**Detailed Comments:**

In the dataset section, the citation of Ambellan 2019 may be inaccurate, the knee data are from the OA initiative (Peterfy et al), while Ambellan 2019 provided the knee segmentation masks?

The shape prior is learned using an auto decoder scheme - a question I would like to ask to the authors is: how many voxels are extracted per shape? If the initial MRI had let’s say 100rows x 100cols x 20slices would some in-plane sampling be done, or all the available voxels are used? - perhaps add a comment to “Note that the number of voxels, as well as voxel sizes, can differ between training volumes.” Explaining what causes the variation.

In my opinion row-col-slices is a better convention when describing the data - this differs from the convention used in some DL frameworks.

Performance is compared against baseline approaches, with appropriate metrics. Showing very good reconstruction performance achieved from various views and orthogonal slices. It would have been nice to report the segmentation performance that a DL model could obtain on the high resolution MRI, I think this would put everything in perspective and potentially impress on how minimal is the degradation in ultimate segmentation quality.

**Final Rating After The Rebuttal:**

5: Strong Accept

**Justification Of The Final Rating:**

I carefully read the response to all the reviewers' comments, and strongly believe this paper is really interesting and the community will benefit from its presentation at MIDL. For this reason I would like to upgrade my rating to strong accept.

**Paper Type:**

methodological development

**Questions To Address In The Rebuttal:**

As part of the rebuttal, I would like the authors to comment, on the points above.

Additionally:
- What are the next steps?
- Do they envision as an ultimate goal not to have the need of training an high res MRI anymore? why would they like that to happen? More High resolution MRI will be at disposal in the future, especially because the image reconstruction community is putting al big effort in obtaining high quality mri with fewer datapoints, .

This method has potential to have an impact in the MSK community, so sharing the code (as promised by the authors) will definitely increase the impact of this paper, as well as understanding of some not so clear steps.


**Special Issue:**

no

---

### Meta-Review · Area_Chair_i7PZ · 2022-02-19

**Recommendation:** Accept (Oral)
**Confidence:** 5

**Metareview:**

The authors propose using neural implicit functions to reconstruct high resolution shapes from low resolution anisotropic, or sparsely sampled segmentation maps.  The approach is based on creating a neural implicit representation of shape, where a shape prior is learned from a training dataset (with the same level of anisotropy/sparseness), but following inference shapes can be generated at arbitrary resolutions. Performance is compared against baseline approaches, with appropriate metrics. Results demonstrate the capability to infer reasonable high resolution shapes from lower-resolution data.

All reviewers agreed that this article presents a potentially exciting avenue for using deep learning in medical shape analysis, and provides concrete example applications for the clinic and the community will benefit from its presentation at MIDL. All reviewers suggested acceptance of the paper. I agree with the reviewers and recommend acceptance.

---

### Decision · Program_Chairs · 2022-02-28

Accept